# Cooperativity of ESPT and Aggregation-Induced Emission Effects—An Experimental and Theoretical Analysis of a 1,3,4-Thiadiazole Derivative

**DOI:** 10.3390/ijms25063352

**Published:** 2024-03-15

**Authors:** Iwona Budziak-Wieczorek, Dominika Kaczmarczyk, Klaudia Rząd, Mariusz Gagoś, Andrzej Stepulak, Beata Myśliwa-Kurdziel, Dariusz Karcz, Karolina Starzak, Gotard Burdziński, Monika Srebro-Hooper, Arkadiusz Matwijczuk

**Affiliations:** 1Department of Chemistry, Faculty of Life Sciences and Biotechnology, University of Life Sciences in Lublin, Akademicka 15, 20-950 Lublin, Poland; iwona.budziak@up.lublin.pl; 2Doctoral School of Exact and Natural Sciences, Jagiellonian University, Prof. St. Łojasiewicza St 11, 30-348 Cracow, Poland; dominika.kaczmarczyk@doctoral.uj.edu.pl; 3Department of Theoretical Chemistry, Faculty of Chemistry, Jagiellonian University, Gronostajowa 2, 30-387 Kraków, Poland; 4Department of Biophysics, Faculty of Environmental Biology, University of Life Sciences in Lublin, Akademicka 13, 20-950 Lublin, Poland; klaudia.rzad@up.lublin.pl; 5Department of Cell Biology, Maria Curie-Sklodowska University, Akademicka 19, 20-033 Lublin, Poland; mariusz.gagos@mail.umcs.pl; 6Department of Biochemistry and Molecular Biology, Medical University of Lublin, Chodźki 1, 20-093 Lublin, Poland; andrzej.stepulak@umlub.pl; 7Department of Plant Physiology and Biochemistry, Faculty of Biochemistry, Biophysics and Biotechnology, Jagiellonian University, Gronostajowa 7, 30-387 Kraków, Poland; b.mysliwa-kurdziel@uj.edu.pl; 8Department of Chemical Technology and Environmental Analytics (C1), Faculty of Chemical Engineering and Technology, Cracow University of Technology, Warszawska 24, 31-155 Kraków, Poland; dariusz.karcz@pk.edu.pl (D.K.); karolina.starzak@pk.edu.pl (K.S.); 9ECOTECH-COMPLEX—Analytical and Programme Centre for Advanced Environmentally-Friendly Technologies, Maria Curie-Sklodowska University, Głęboka 39, 20-033 Lublin, Poland; 10Faculty of Physics, Adam Mickiewicz University, Poznań, Uniwersytetu Poznańskiego 2, 61-614 Poznań, Poland; gotardb@amu.edu.pl

**Keywords:** 1,3,4-thiadiazole, excited-state proton transfer (ESPT)/excited-state intramolecular proton transfer (ESIPT), aggregation-induced emission (AIE), twisted intramolecular charge transfer (TICT), dual fluorescence, molecular spectroscopy, density functional calculations

## Abstract

*4-[5-(Naphthalen-1-ylmethyl)-1,3,4-thiadiazol-2-yl]benzene-1,3-diol* (NTBD) was extensively studied through stationary UV–vis absorption and fluorescence measurements in various solvents and solvent mixtures and by first-principles quantum chemical calculations. It was observed that while in polar solvents (e.g., methanol) only a single emission band emerged; the analyzed 1,3,4-thiadiazole derivative was capable of producing dual fluorescence signals in low polarity solvents (e.g., *n*-hexane) and certain solvent mixtures (e.g., methanol/water). As clearly follows from the experimental spectroscopic studies and theoretical modeling, the specific emission characteristic of NTBD is triggered by the effect of enol → keto excited-state intramolecular proton transfer (ESIPT) that in the case of solvent mixture is reinforced by aggregation of thiadiazole molecules. Specifically, the restriction of intramolecular rotation (RIR) due to environmental hindrance suppresses the formation of non-emissive twisted intramolecular charge transfer (TICT) excited keto* states. As a result, this particular thiadiazole derivative is capable of simultaneously producing both ESIPT and aggregation-induced emission (AIE).

## 1. Introduction

As medicine continues to struggle with various ramifications of cancer and neurodegenerative diseases [1], scientists are making efforts to identify and test new compounds whose specific molecular properties may potentially facilitate desired therapeutic effects. However, since the exact mechanisms responsible for the biological and pharmacological activity of such systems are often unclear, it is difficult to identify possible ways to fine-tune/modify them. It is also often the case that pre-clinical and clinical studies fail to include spectroscopic characterization of newly introduced medicines. As such, the potential of fluorescence-based methods is rarely fully realized, despite their exceptional sensitivity.

A number of 1,3,4-thiadiazole derivatives have been shown to offer promising antimicrobial, antitumor, or neuroprotective properties [2,3,4,5,6], and several have already entered clinical trials or found practical therapeutic applications [7]. Numerous 1,3,4-thiadiazole-based molecules have also been considered in the process of designing novel lasers [8], oxidation inhibitors [9], and chelating agents for complexation of various d-block metal ions [10]. In recent years, many research teams, ours included, have been engaged in studies of 1,3,4-thiadiazole derivatives demonstrating a variety of interesting crystallographic, spectroscopic and biophysical properties [11], often considered as essential for pharmacological potential. Some of the identified spectroscopic effects include changes in spectral characteristics in complexes with transition metal ions [12], dynamics changes in lipid systems [13,14], or the emergence of dual fluorescence. Rationalization of the latter is usually based on the coexistence of multiple emissive states in different electronic and/or molecular structures, which is associated with the processes of excited-state proton transfer (ESPT) [15] and twisted intramolecular charge transfer (TICT) [16,17] or breaching the Kasha’s rule [18], as well as aggregation effects related to the formation of excimer systems [19,20,21] and aggregation-induced emission (AIE) or aggregation-induced emission enhancement (AIEE), typically attributed to restriction of intramolecular rotation (RIR) [22]. With regard to the particular group of 2,4-dihydroxyphenyl-substituted 1,3,4-thiadiazoles, dual fluorescence has been most commonly explained by referring to ESPT effects, more specifically excited-state intramolecular proton transfer (ESIPT) [23,24,25], i.e., a proton transfer occurring within the molecule itself, as well as aggregation [26,27]. We believe that a more in-depth examination of the mechanisms governing these spectroscopic phenomena could facilitate a better understanding of how the specific structural features influence the pharmacological activity of 1,3,4-thiadiazoles.

If the proton donor group (e.g., phenolic OH) is located near the proton acceptor group (e.g., carbonyl oxygen or the nitrogen atom in the heterocyclic ring), both connected by an intramolecular hydrogen bond, photoexcitation may affect the electron density distribution, resulting in an increase in the acidity and basicity of the respective sites, which in turn can induce the ultrafast ESIPT phototautomerization process [23,28]. Typical markers of the described effect include extensive Stokes shifts, specific multi-emission spectra, and increased sensitivity to the type of medium used [23,29]. Regarding the latter, it was shown that increasing the solvent’s polarity and its ability to form hydrogen bonds with the solute (which may disrupt pre-existing intramolecular hydrogen bonding along which ESPT takes place) can rapidly reduce ESIPT capacity [30]. ESIPT processes have been under considerable scientific scrutiny for several decades [31], contributing to numerous practical applications, such as laser pigments [32], solar panels [33,34], light-emitting diodes (LEDs) [35], sensors [36,37,38], and UV light polymer stabilizers [39]. ESIPT-capable molecules are also increasingly utilized as molecular probes in protein and DNA-related systems [40,41]. Interestingly, recent studies have demonstrated the existence of a new, sequential type of proton transfer, so-called proton transfer triggered proton transfer (PTTPT), in which the original ESIPT triggers a secondary ESIPT [42], which paves the way for new potential functions of ESPT-type systems.

The study presented below explored the mechanisms of molecular fluorescence effects occurring in a 1,3,4-thiadiazole derivative with verified pharmacological properties (*4-[5-(naphthalen-1-ylmethyl)-1,3,4-thiadiazol-2-yl]benzene-1,3-diol*—NTBD, Figure 1) [43,44]. The structure of NTBD is conducive to intramolecular hydrogen bonding between the proton donor (OH group in the resorcynyl ring) and proton acceptor (N atom of the 1,3,4-thiadiazole) groups, which is a structural requirement for tautomerization involving ESIPT. As a follow up to our earlier study focused primarily on examining the spectroscopic properties of NTBD in DMSO/water solutions [44], the present article details further comprehensive examination of this system in several other solvents of varying polarity and methanol/water mixtures using a variety of spectroscopic techniques, including stationary absorption spectroscopy, fluorescence spectroscopy (with resonance light scattering, RLS), and time-resolved emission lifetime measurements in the frequency domain. The determined quantum yields and rate constants for radiative and non-radiative processes in NTBD molecules in both pure and mixed media were analyzed, and for the former cases (pure solvents) these measurements were related to changes in the polarity function E_T_(30). Equally important, thorough first-principles (time-dependent) density functional theory ((TD)DFT) computations involving modeling of ESIPT and formations of TICT state pathways were conducted to corroborate and rationalize the experimental findings. The combined experimental and theoretical analysis evidenced cooperativity between excited-state proton transfer and molecular aggregation processes (which prevent the occurrence of non-fluorescent TICT states via the RIR mechanism) responsible for the (dual) emission characteristics of NTBD.

## 2. Results and Discussion

### 2.1. Structural Preferences Analysis

Figure 1 illustrates the structure of NTBD. The molecule is composed of a resorcynyl ring bound with the 1,3,4-thiadiazole moiety, with the -CH_2_- group additionally linking it to a naphthalene fragment (mainly determining the polarity and phase division of the studied system) [44]. More importantly, however, the hydrogen atom of the resorcynyl *ortho* OH group in NTBD may be positioned in close proximity to the 1,3,4-thiadiazole nitrogen atom. Such arrangement may promote the formation of an intramolecular hydrogen bond between the two sites (as in the *cis*-enol structure, Figure 1a) and consequently enol–keto tautomerization, rendering the compound an ideal system to consider in a study on ESIPT effects. Moreover, an NTBD enolic structure is also possible with the hydrogen atom of the resorcynyl *ortho* OH group pointing towards the *para* OH (*trans*-enol in Figure 1a), which would preclude said hydrogen bonding.

The selected low-energy DFT-optimized NTBD structures obtained for each of the isomers shown schematically in Figure 1a, namely *cis*-enol, keto, and *trans*-enol, are presented in Figure 1 and Appendix A; see also Appendix A for quantitative characterization of the examined structures. All the considered geometries demonstrate the planar arrangement of the resorcynyl and 1,3,4-thiadiazole rings, and within each of the isomeric group they differ only in the position of the –CH_2_-naphthalene substituent with respect to the resorcynyl–thiadiazole fragment. It should also be noted that the corresponding structures, differing only in the position of the hydrogen atom of the *para* OH group in the resorcynyl fragment (labelled without and with the prime symbol, see Appendix A), demonstrate almost identical structural parameters and energy values. As expected, the low-energy *cis*-enol and keto structures are stabilized by an intramolecular hydrogen bonding, O–H···N and O···H–N, respectively, between the resorcynyl and thiadiazole moieties, whereas in the *trans*-enol geometries, the resorcynyl ring is rotated around the CC bond connecting it to the thiadiazole such that its *ortho* OH group is positioned on the side of the sulfur atom of the latter fragment and the hydrogen atom of this group points in the opposite direction than the sulfur, so no O–H···S hydrogen bond was formed (Appendix A) [45]. More importantly, in agreement with the experimental data (see UV–vis absorption analysis below, Figure 2 and Figure 3), the computational results confirm that in the polar (such as methanol) solution the NTBD system should exist preferentially in the *cis*-enol form, as the *trans*-enol and keto structures have significantly higher and comparable relative energy and free energy values (>ca. 5.5 kcal/mol) [44]. The predominant population of this tautomer in polar solvents appears to be also supported by computations of the energy profiles for the *cis*-enol ↔ keto tautomerization process for NTBD (Appendix A). As presented in Figure 1 and Appendix A, the estimated energy barrier for the *cis*-enol → keto transition is ca. 7.6 kcal/mol, while the reverse (keto → *cis*-enol) transformation is essentially barrierless (<1 kcal/mol). This indicates that even if the keto tautomer is formed, a fast re-isomerization leading to the enol tautomer likely occurs.

### 2.2. Studies of Electronic Absorption in Pure Solvents

The results of spectroscopic measurements of electronic absorption conducted for NTBD in a number of solvents with varying polarity are presented in Figure 2a. The energetic positions of absorption maxima and physical constants for all the solvents considered are provided in Appendix A. Two relatively wide bands are evident in the presented UV–vis spectra: the higher-energy one between 260 and 300 nm with the maximum at ca. 280 nm, and the lower-energy one between 300 and 360 nm with the maximum at ca. 320 nm. With decreasing solvent polarity, a bathochromic shift of Δλ = 6 nm (575 cm^−1^) emerged for the main (lower-energy) band from 320 nm to 326 nm (Figure 2a and Appendix A). The slight but noticeable shift of this main absorption maximum is a clear indication of the compound’s sensitivity to changes in medium polarity. In some solvents, an additional relatively low-intensity band emerged on the long-wavelength side of the spectrum, with the maximum at ca. 375 nm.

The TDDFT-simulated UV–vis absorption spectra for *cis*-enol 1 and 2 and their corresponding keto 1 and 2 NTBD structures are presented in Figure 3; see Appendix A for the spectra obtained for all the computationally considered geometries of NTBD. As can be seen, each type of examined structure (*cis*-enol, keto, *trans*-enol) exhibits quite distinct spectral envelopes that are only weakly dependent on the position of the –CH_2_-naphtalene group with respect to the resorcynyl–thiadiazole fragment and the position of the hydrogen atom of the *para* OH group in the resorcynyl moiety. The UV–vis spectra obtained for the lowest-energy structures, i.e., *cis*-enol, display two bands with shapes and wavelengths of absorption maxima consistent with those physically measured, although some deficiencies in the reproduction of their relative intensity are observed (Figure 2a and Figure 3). The spectra computed for the keto structures also show two bands in the considered spectral region, but their envelopes and energetic position significantly deviate from the experimental ones—the simulations led to two well-separated bands with the lower-energy one tremendously bathochromically shifted (red-shifted) compared to the main experimental intensity. Finally, the *trans*-enol structures demonstrate spectra resembling those computed for the *cis*-enol geometries, but hypsochromically shifted (blue-shifted) with respect to the experiment and with higher-energy bands showing inadequately low intensity (Appendix A). A good agreement between experimental and simulated spectra for the *cis*-enol NTBD structures confirms further the dominance of this tautomer in polar solvents such as methanol. However, we noted in passing that the experimental UV–vis signal around 375 nm observed in some solvents agrees well with the position of the low-energy band computed for the keto NTBD geometries, which may indicate a non-zero population of this tautomer in such media. Nevertheless, the intensity of this band, as well as the extent of its bathochromic shift, may also be indicative of some presence of aggregated forms in the analyzed compound (e.g., dimers, higher-order aggregates, see Figure 1b) [46,47].

Further analysis of the dominant excitations computed for the NTBD *cis*-enol structures showed that the main (lower-energy) band observed at around 320 nm originates from S_0_ → S_1_ and/or S_0_ → S_2_ electronic excitations involving predominantly (highest occupied molecular orbital) HOMO → (lowest unoccupied molecular orbital) LUMO and HOMO-1 → LUMO transitions that collectively correspond to the π-π* transition within the resorcynyl–thiadiazole π-electron system and the naphthalene → resorcynyl–thiadiazole charge transfer (CT) [48] (see Figure 3, Appendix A). Importantly, as revealed by the results obtained, for example, for the *cis*-enol 1 structure, both resorcynyl–thiadiazole-centered π-π* (described via HOMO → LUMO) and naphthalene → resorcynyl–thiadiazole CT (represented by HOMO-1 → LUMO) transitions almost equally contribute to this band, as they both are involved in isoenergetic excitations no. 1 and 2 but with reverse percentages (π-π*/CT: 62%/34% (no. 1) and 32%/65% (no. 2)). The higher-energy UV–vis intensity measured at around 280 nm is attributed mainly to S_0_ → S_4_ and S_0_ → S_5_ excitations that involve HOMO-2, HOMO-1, LUMO, and LUMO+1 and can be assigned as an admixture of the π-π* transition within the naphthalene unit and CT-like π-π* within resorcynyl–thiadiazole. The dominant excitations responsible for the corresponding bands in the simulated spectra of the keto structures demonstrate similar character and origin to those found in the *cis*-enol isomers—π-π* within the resorcynyl–thiadiazole for the low-energy band and within both the naphthalene and resorcynyl–thiadiazole for the high-energy band—but without CT contributions. Finally, an inspection of the corresponding MOs for the *cis*-enol and keto forms (see Figure 3 and Appendix A) revealed that the *cis*-enol → keto tautomerization hardly affects distribution of the electron density across the structure apart from—as expected—the resorcynyl–thiadiazole π-electron system and has a strongly stabilizing effect on the LUMO and an even more strongly destabilizing effect on the HOMO of keto. This leads to a decrease in the HOMO–LUMO gap for keto vs. *cis*-enol, which may rationalize the very large bathochromic shift of the low-energy band in the former vs. latter isomers observed in the modeled spectra.

### 2.3. Fluorescence Emission and Resonance Light Scattering Measurements in Pure Solvents

Stationary fluorescence spectroscopy results obtained for highly diluted samples of NTBD (~10^−5^ M, in order to avoid unwanted interference related to inner filter effects [49]) are shown in Figure 2b alongside corresponding absorption spectra given in panel a; the numerical emission data for all the examined solvents are provided in Appendix A. As follows from the graphs (and in agreement with our earlier study [44]), depending on the solvent polarity, a single excitation triggered either a typical single fluorescence emission band or a dual signal. Specifically, in MeOH and DMF (as well as other polar solvents listed in Appendix A), excitation at the wavelength corresponding to the main absorption maximum (associated with the resorcynyl–thiadiazole π-π* and naphthalene → resorcynyl–thiadiazole CT transitions within the *cis*-enol form, vide supra) yielded a single emission centered at approximately 375 nm. In lower-polarity solvents (e.g., *n*-hexane), dual fluorescence emerged with the maximum of the first, higher-energy (shorter-wavelength) band at approximately 370–410 nm and another, lower-energy (longer-wavelength) band at around 500 nm. The fact that an extensive Stokes shift was observed for the low-energy emission indicates a substantial change in the molecular structure of the underlying excited state, likely related to *cis*-enol → keto ESPT [23,50] and/or aggregation effects [51] (vide infra). It should be noted that the intensity of the single emission band was visibly higher than that of the double fluorescence signals. This indicates a significantly lower fluorescence quantum yield of a specific molecular form responsible for the effect in question and/or the existence of an effective emission quenching pathway (vide infra). Moreover, it is noteworthy that observable differences in the emission spectra of the compound in ACN were registered depending on the excitation wavelength (Figure 2b). This may suggest that molecule-to-solvent proton transfer can also occur and that relaxation channels are affected, among other factors, by the nature of the solvent.

Resonance light scattering (RLS) measurements were then performed to verify the occurrence of molecular aggregation for NTBD in pure solvents. As reported in the literature [52], the emergence of RLS bands should be associated with the presence of molecular chromophoric aggregates in the studied solution. The RLS spectra measured for NTBD in several solvents of varying polarity are presented in Figure 4; the corresponding changes in RLS intensity at 386 nm relative to the medium dielectric constant are presented in the insert. It is evident that strong RLS spectra emerged mainly in solvents in which no dual emission occurred, while in the solvents in which the latter was observed, the intensity of RLS bands was significantly lower, albeit noticeable at higher concentrations. Notably, the fact that the recorded RLS spectra showed an oscillatory structure indicates that various associative forms of NTBD exist, which can likely be associated with the extent of aggregation. All this indicates that ESIPT is more likely than molecular aggregation to be the key factor responsible for the dual fluorescence effect observed for NTBD in pure solvents. Accordingly, based on the literature [31] and as established herein via conducted quantum chemical calculations (vide infra), the higher-energy band of the signal in question corresponds to the locally excited enol* form, and the lower-energy one to the excited keto* tautomer, which, following the emission, re-isomerizes to the more stable in S_0_ enol state via a fast proton transfer (vide supra).

### 2.4. Experimental Studies of Photophysical Features in Methanol/Water Solutions

Remarkably, in the course of further examination, it was observed that, unlike in pure solvents, the photophysical behavior of NTBD in a mixed system (in this work, methanol/water; see also results for DMSO/water in [44]) suggests a possible interplay between ESPT and aggregation processes. Figure 5 presents the corresponding electronic absorption and emission spectra for NTBD in MeOH and H_2_O solutions mixed at various ratios. As indicated in panel a, in mixtures in which the contribution of MeOH was higher than or comparable with that of H_2_O, a typical NTBD absorption spectrum was observed with the main (lower-energy) maximum centered at approximately 320 nm and identical to that measured in the pure methanolic NTBD solution (compare with Figure 2a). However, when the MeOH:H_2_O ratio reached ca. 2:8, and very clearly at 1:9, the primary maximum was bathochromically shifted from 320 nm to ca. 327 nm, and as far as 330 nm at the ratio of 1:99. This was accompanied by a general decrease in absorption intensity for wavelengths shorter than ca. 340 nm and a significant enhancement on the longwave side of the spectrum. Kasha’s exciton splitting theory associates such electronic absorption spectrum changes with the influence of chromophoric aggregation in a molecular system, with the described long-wavelength shift indicative primarily of the formation of (offset) “card pack” aggregates, e.g., the π-π-stacked forms presented in Figure 1b [51,53]. In our previous publications, we demonstrated the readiness of thiadiazole derivative molecules to aggregate in the presence of water as a result of significantly strengthened hydrophobic interactions [54]. As can be inferred from Figure 5b presenting the ratio of NTBD absorption intensity measured at 320 nm and at 330 nm relative to the MeOH:H_2_O volume ratio, the higher the content of water in the sample, the more pronounced the aggregation effects become. Their influence is the clearest in the 1:99 MeOH/H_2_O mixture, for which the full width at half maximum (FWHM) parameter of the main absorption spectrum band is additionally significantly enhanced (from 317 nm to 330 nm), which further confirms that aggregation indeed takes place.

As shown in Figure 5c, the emission spectra of NTBD in MeOH/H_2_O solutions are considerably more sensitive to changes in system composition than the corresponding absorption spectra. More specifically, when even a small amount of water was added (MeOH:H_2_O of 8:2), dual fluorescence was present with the maxima at ca. 380 nm and 440 nm. With increasing water content, the higher-energy band quickly diminished while the lower-energy band remained relatively intense. In solutions with predominance of H_2_O (MeOH:H_2_O of 1:9 and 1:99), the higher-energy band practically disappeared and the low-energy band was replaced by a longwave signal centered at ca. 500 nm, similar to that observed for NTBD in pure, low-polarity solvents such as *n*-hexane (see Figure 2b). A quantitative analysis of the intensity changes observed for the NTBD emission bands obtained at various MeOH:H_2_O ratios is presented in Figure 5d. Comparing the fluorescence spectra measured for NTBD in pure solvents and in mixed MeOH/H_2_O (and DMSO/H_2_O [44]) solutions, it can be concluded that the dual fluorescence effect appearing in the latter is very likely due to ESPT (with the high-/low-energy signal indicative of enol*/keto* tautomer), but not without a non-negligible contribution from aggregation processes (vide infra), the presence of which was further corroborated by the NTBD fluorescence excitation and RLS spectra recorded in MeOH/H_2_O mixtures.

Higher selectivity of excitation spectra when compared to electronic absorption makes the effects related to molecular aggregation much more easily visible in the former. As seen in Appendix A, even with a tiny addition of water into the methanol solution of NTBD, there was a noticeable shift in the primary maximum in the excitation spectrum from ca. 320 nm in the bathochromic direction, as well as a substantial widening of the band on the longwave side and an increase in its FWHM. At the MeOH:H_2_O ratio of 1:9 or 1:99, the spectral shift was very significant, to ca. 336 nm (1:9), and accompanied by a simultaneous intensity reduction (1:99), which in this particular case was due to extensive aggregation. Appendix A presents a juxtaposition of the corresponding excitation (Ex) and 1-T (T = transmission) spectra for two selected MeOH:H_2_O ratios (1:9 and 2:8), with each panel showing also the corresponding differential Ex–(1-T) spectrum. In panel b, presenting the data for the 2:8 ratio, the contribution from aggregation is clearly evident in the differential spectrum, far more so than in the excitation spectra alone. Finally, Appendix A illustrates respective RLS spectra corresponding to the fluorescence excitation spectra measured for NTBD at various MeOH:H_2_O ratios. It can be noted that the RLS signal was already clearly detectable at the ratio of 4:6, which indicates strong aggregation effects that are far more evident than in a pure MeOH solution of NTBD (compare with Figure 4).

Figure 6 and Appendix A present the results of additional experiments conducted to complement the primary examination of NTBD aggregation. In particular, the impact of temperature on NTBD aggregation effects was examined in a 5:5 MeOH/H_2_O solution. The fluorescence emission spectra registered in the selected temperature range are shown in Figure 6. Based on the obtained data, one can observe that an increase in temperature triggered a significant decrease in the intensity of the lower-energy band, while as temperature decreased, the band intensity returned to its original magnitude. This effect of the proportional change in intensity between the emission bands with the maxima at ca. 380 nm and ca. 440 nm may suggest hysteresis-like behavior but is in fact fully reversible (see Figure 6c), which evidences high stability in the analyzed system.

The electronic absorption spectra and the corresponding fluorescence emission spectra for increasing amounts of NTBD dissolved in a 5:5 MeOH/H_2_O mixture are shown, respectively, in panels a and b of Appendix A. While the UV–vis spectra showed no significant changes apart from a typical absorbance enhancement expected with the increasing concentration of the compound, the emission signal revealed a slight enhancement in the lower-energy band with the maximum at ca. 435 nm upon rising NTBD content in the mixture, which suggests that aggregation processes primarily affect the band associated with the keto* tautomer. The excitation spectra presented in Appendix A measured at wavelengths corresponding to the fluorescence emission maxima shown in Appendix A clearly demonstrate the tendency of NTBD to aggregate under such conditions. As can be seen, with increasing concentration of the compound, the primary lower-energy band was significantly broadened as well as enhanced on the long-wavelength side. The conclusion following from these results is that under appropriate measurement conditions, aggregation effects may indeed facilitate emission from the keto* form.

### 2.5. Time-Resolved Fluorescence Spectroscopy Studies in Pure and Mixed Systems

Evidence to support the ESPT-based origin of dual fluorescence in NTBD was eventually provided by measurements of fluorescence lifetimes performed in selected solvents at low (ca. 1 × 10^−5^ M) and high (ca. 1 × 10^−3^ M) NTBD concentrations, as well as in MeOH/H_2_O mixtures. Data obtained using a 320 nm filter are presented in Figure 7 and Figure 8 and Table 1; see Appendix A for results recorded with a 305 nm filter. Frequency domain measurements revealed that in most cases two distinct lifetimes were clearly identifiable, which confirms the co-existence of two different emitting NTBD states. Time-resolved fluorescence spectroscopy is a highly sensitive and specific experimental method that facilitates detection of different tautomeric forms of a fluorophore in polar and non-polar environments [55,56]. As presented in Figure 7a, in solvents wherein a single emission of NTBD took place, only the long-life component of 1.5–2 ns was present, while in cases in which fluorescence lifetime was registered at longer wavelengths, i.e., in clear cases of dual fluorescence, the observed decay was distinctly binary with the additional shorter lifetime of ca. 0.5 ns. According to the literature [55], the excited keto* form has been associated with this shorter lifetime, while the longer lifetime has been linked to the excited enol* tautomer. Interestingly, although the short fluorescence lifetime component was not observed in the case of low NTBD concentrations in either MeOH or DMF, indicating the predominance of enol* forms under such conditions, it was detected upon an increase in the amount of the compound in the sample (Figure 7c), which suggests activation of the effective ESIPT process in higher NTBD concentrations. Note that the corresponding fractions of the respective lifetimes follow a similar pattern (Figure 7b,d). Finally, fluorescence lifetimes registered for NTBD in varying ratio MeOH/H_2_O solutions are presented in Figure 8 to more clearly showcase the interplay between ESIPT and aggregation effects. As shown in the graph, the higher the content of H_2_O in the mixture, the greater the decrease in the mean lifetime. Two constituents could, however, be identified for some of the ratios, which corresponds well with the presence of dual fluorescence in such mixed NTBD systems.

### 2.6. Quantum Chemical Analysis of Fluorescence Emission

To shed a light on experimentally observed emission features of NTBD in both pure and mixed solutions, and, in particular, to ultimately assign an origin of the dual fluorescence signals and understand the impact of aggregation effects on thereof, quantum chemical studies were then performed. The calculations (TDDFT B3LYP/aug-cc-pVDZ with the continuum solvent model for MeOH and for H_2_O) involved S_1_ excited-state geometry optimizations of NTBD in its two tautomeric forms, that is enolic (as both *cis*-enol and *trans*-enol) and keto, and the corresponding S_1_ energy profile computations for the intramolecular *cis*-enol ↔ keto proton transfer and for the rotation of the resorcynyl fragment relative to the 1,3,4-thiadiazole unit, representing ESIPT and TICT processes, respectively. The selected results are presented in Figure 1 (which simultaneously summarizes the possible S_1_ → S_0_ energy transfer route) and Figure 9, while a full set of calculated data can be found in Appendix A.

As can be seen in Figure 1 and Appendix A (see also Appendix A for quantitative characterization of all the optimized S_1_ structures), the excited states obtained via direct optimization of the ground-state NTBD geometries clearly resemble their S_0_ ‘parent’ structures with the respective enol or keto character and planar resorcynyl–thiadiazole conformation preserved. As far as energetic preferences of these structures are concerned, for *cis*-enol and keto forms a striking difference between S_0_ and S_1_ occurs. Namely, unlike for S_0_, the excited keto* structures demonstrate significantly lower energy and free energy values (by ca. 6–7.5 kcal/mol) than the excited *cis*-enol*. Note that, as in S_0_, the *trans*-enol structures also remain strongly energetically disfavored in S_1_ (by ca. 7–11 kcal/mol as compared to *cis*-enol* and ca. 13–17 kcal/mol as compared to keto*).

Importantly, the calculations also revealed that all these conformers are emissive, and the corresponding fluorescence characteristics clearly depend on the type of tautomeric form. Namely, both considered enol structures—*cis*-enol* and *trans*-enol*—demonstrate rather similar S_1_ → S_0_ fluorescence wavelengths, ca. 380–400 nm depending on the geometry, corresponding to LUMO → HOMO π-π* transition within resorcynyl–thiadiazole fragments with an involvement of π-electron systems of the naphthalene unit in some cases. Such pure π-π* assignment affords sizable values of oscillator strength *f* that would translate into high emission intensity (see Figure 9, and Appendix A). Regarding the keto* tautomer, the computations showed that its emission occurs at bathochromically shifted wavelengths (ca. 470 nm) and with decreased intensity (diminished *f*) compared to enol*. Both features can be traced back to the (even more pronounced than in S_0_) impact of the tautomerization process on the resorcynyl–thiadiazole π-electron system (see isosurfaces of HOMO in Figure 9 and Appendix A, compare with Figure 3), leading to a clear CT-like character of the LUMO → HOMO π-π* transition within resorcynyl–thiadiazole fragments in keto* and a reduction in the HOMO–LUMO gap for keto* vs. *cis*-enol* due to a strong destabilizing/stabilizing effect on the HOMO/LUMO of keto*.

Taking into account the visibly higher energy of *trans*-enol structures in both S_0_ and S_1_ as compared to *cis*-enol, which renders the respective *cis*-enol → *trans*-enol isomerization as highly improbable in the ground state as in the excited state, the contribution of *trans*-enol* to the emission of NTBD seems negligible. Importantly, subsequent computations of the energy profiles for the *cis*-enol ↔ keto tautomerization process for NTBD in S_1_ (Figure 1, Appendix A) revealed, however, a significantly decreased energy barrier for the *cis*-enol* → keto* transition (ca. 2–3.5 kcal/mol) and a significantly increased barrier for the reverse keto* → *cis*-enol* transformation (ca. 8–9 kcal/mol) relative to S_0_ (vide supra). The changes in the barriers clearly correlate with the decreased/increased respective interatomic OH···N/O···HN distance in *cis*-enol/keto structures of S_1_ vs. S_0_ (see Figure 1 and Appendix A). Accordingly, within the computational approach used in this work, ESIPT enol → keto tautomerization of NTBD in its excited state in pure solvents seems to be indeed both kinetically and thermodynamically allowed.

Interestingly, the calculations demonstrated also that an intramolecular rotation of the resorcynyl group (relative to the thiadiazole unit) in the aforementioned S_1_ keto form may easily (with the energy barrier < 1 kcal/mol) lead to other stable minima on the excited-state potential energy surface (PES) of NTBD that show a significant deviation from planarity of both fragments (ca. 112°) and are characterized by even lower energy and free energy values than their ‘parent’ planar structures (by ca. 4–7 kcal/mol) (see keto* structures labelled with the superscript ‘twist’ in Figure 1, Appendix A). Due to their non-planar structure and the resulting loss of the π-conjugation across the resorcynyl–thiadiazole electron systems, such conformers reveal a non-fluorescent TICT character, with the corresponding LUMO → HOMO emission transition representing charge transfer between mutually orthogonal π-electronic units, namely the thiadiazole (LUMO) and resorcynyl (HOMO), that affords no intensity (see Figure 9 and Appendix A). This suggests that after the formation of keto* structures via ESIPT, non-radiative deactivation channels for NTBD in solution may be easily opened, likely involving the TICT states. It is also worth noticing that the formation of analogous non-emissive ‘twist’ NTBD conformers of *cis*-enol* was shown to be highly improbable due to the high values of the corresponding rotation energy barriers (ca. 10–12 kcal/mol) and the high values of the relative energies and free energies of such structures (by ca. 8.5–12 kcal/mol with respect to their planar ‘parent’ forms), ensuring low barriers for the reverse transformation (Figure 1 and Appendix A). Furthermore, even though such a rotation would take place, the resulting ‘twist’ structure could easily (energy barrier < 1 kcal/mol) move to a nearby ‘less twisted’ minimum on the *cis*-enol* PES that demonstrates emissive character similar to that described above for the planar enol* conformers (see structures labelled as ‘twist-Ib’ in Figure 1). Additionally, since such rotation of the resorcynyl ring relative to the thiadiazole fragment is also required for isomerization between *cis*-enol and *trans*-enol structures, its high-energy barrier further supports a negligible population of the latter in the excited state of the NTBD compound.

The satisfactory agreement between computed S_1_ → S_0_ emission wavelengths and the experimentally measured energetic positions of the single fluorescence emission band and higher-energy band of the dual fluorescence signal (in the case of *cis*-enol*), and of the lower-energy band of the dual fluorescence signal (in the case of keto*), ultimately confirms the assignments of these signals. Moreover, it clearly evidences that predominance of the former NTBD tautomer in a solution leads to the single emission, while the dual fluorescence effect is due to co-existence of both tautomers in S_1_ as a result of either their co-existence in S_0_ or the efficient ESIPT process from the locally excited *cis*-enol* to the corresponding keto* that, after emission, quickly isomerizes to the more stable in S_0_ enol state.

Regarding the NTBD in pure solvents, the occurrence of such phototautomerization in the excited state can indeed account for the dual emission observed in non-polar media such as *n*-hexane. It is worth mentioning that due to a possibly non-negligible population of keto tautomer in the ground state in such solvents, the excited keto* may be formed also as a locally excited state, similarly to *cis*-enol* [57]. Furthermore, the low intensity of the emission signals in these cases (Figure 2b) is fully consistent with not only relatively low values of computed oscillator strength for the planar keto* structures (vs. those for *cis*-enol*), but primarily a feasible formation of ‘twist’ keto* conformers of non-emissive TICT character, which results in an additional shift in the equilibrium between planar *cis*-enol* and the corresponding keto* towards the latter, further reinforcing emission quenching. This can be traced back to the rather negligible aggregation of NTBD in such solvents, as shown by the respective low RLS intensity (vide supra, Figure 4), and thus no significant restrictions on intramolecular rotations in the molecule. The occurrence of only one strongly intense emission band of NTBD in polar solvents such as MeOH is, on the other hand, consistent with predominance of the *cis*-enol form and no presence of the keto tautomer of the compound in S_1_. This may appear in striking opposition to the low values of the computed energy barrier for ESIPT in MeOH that would indicate that the process in such a medium is feasible. It is thus possible that the computational model adopted in these studies is not accurate enough, especially as far as description of environmental effects is concerned. In a protic medium such as MeOH, the intermolecular hydrogen bonding interactions between the solute and the solvent (impossible to simulate within the employed continuum solvent model) may indeed perturb the intramolecular O–H···N bonding in the *cis*-enol, obstructing ESIPT. The dimerization process of NTBD with a hypothetical formation of species presented in Figure 1b (left side) may have the same hindering effect on appearance of the excited keto*, as in such a case a synchronous transfer of two protons, either within each molecule separately or between two molecules, is required for phototautomerization. This is expected to show a higher energy barrier compared to a standard ESIPT process within a single molecule. It is imaginable that such dimeric species may dominate among aggregates of NTBD in MeOH at low concentration, aggregation of which under such conditions was indeed confirmed via RLS measurements (vide supra, Figure 4). Upon an increase in the compound concentration, the equilibrium between single solvated NTBD molecules or their dimers and higher-order aggregates is expected to be shifted towards (offset) “card pack” species (the right side of Figure 1b), for which the intramolecular *cis*-enol* → keto* phototautomerization seems to follow a single-molecule ESIPT and thus becomes feasible. Accordingly, the keto* tautomer may be formed and consequently may contribute to the emission of the system. This is indeed confirmed by the observation of two characteristic lifetimes under such conditions (vide supra, Figure 7 and Table 1).

Undoubtedly, the co-existence of both *cis*-enol* and keto* forms of NTBD upon its excitation in the mixed MeOH/H_2_O solution is consistent with the dual fluorescence signal observed in such media. As aforementioned, the addition of water into MeOH solution of NTBD enhances the aggregation of the compound with a predominant formation of (offset) “card pack” aggregates. Furthermore, it is also known that such a solvent may facilitate tautomerization processes via so-called water-assisted proton transfers both in ground and excited states, as well as in aggregated structures [58,59,60]. All these facts support the hypothesis that both tautomers contribute to the emission signal for NTBD in the mixed MeOH/H_2_O solution, with the keto* species formed as a locally excited state (due to a non-negligible population of such an isomer in the ground state) or due to efficient ESIPT processes. Moreover, unlike NTBD in non-polar solvents, such as for example *n*-hexane, the (offset) “card pack” aggregation of the compound in MeOH/H_2_O mixtures enhances radiative emission pathways from keto* due to enforcing the co-planarity of resorcynyl and thiadiazole rings in its molecular structure (via the RIR mechanism) and, accordingly, due to elimination of the formation of non-emissive TICT keto* states [61,62]. This indeed agrees well with the sizable intensity of the lower-energy band of the keto* origin observed in the experiments (Figure 5c). Finally, it is also worth noticing that the lower intensity of this band compared to that assigned to enol* is consistent with the lower oscillator strength values computed for its underlying electronic transition. Note that the similarity between experimental emission features of NTBD observed in DMSO/H_2_O [44] and MeOH/H_2_O (this work) mixtures clearly supports the crucial role of water molecules in the appearance of dual fluorescence signals for this system.

While the numerical values of the calculated results should be treated rather cautiously due to the possibly limited accuracy of the employed computational approach (standard TDDFT with continuum solvent model), their analysis in relation to the experimental data allowed us, as presented above, to propose a comprehensive rational mechanism underlying the photophysics of NTBD in different solvent media. However, it should be highlighted that in order to fully demonstrate the impact of solvation and/or aggregation effects on the emission behavior of the examined system, further computational research, including molecular dynamic simulations, would be needed, which is beyond the scope of this already extensive work.

### 2.7. Analysis of Stokes Shifts, Fluorescence Quantum Yields, and Radiative and Non-Radiative Decay Constants

To complement the experimental studies presented so far, the spectroscopic results were analyzed by calculating the corresponding values of fluorescence quantum yield Φ_F_ as well as radiative k_r_ and non-radiative k_nr_ decay constants for NTBD in respective pure solvents [44] and in MeOH/H_2_O mixtures. Note that for the observed instances of dual fluorescence, these values quantify the whole emission process. Table 2 and Table 3, as well as Figure 10 and Figure 11, present the results of said calculations. The quantum yield values obtained for NTBD in selected solvents against the corresponding dielectric constants ε were previously analyzed in [44]. It can be observed that the quantum yield Φ_F_ is significantly lower for solvents in which fluorescence emission spectra evidenced the presence of ESIPT effects (ca. 0.1) as compared to solvents that primarily facilitated enol fluorescence (up to ca. 0.7). Changes in NTBD Φ_F_ in MeOH/H_2_O solutions depending on the mixture’s volume ratio are presented in Figure 10. The results indicate a significant increase in Φ_F_ values in cases in which the enol*-origin single emission band was dominant. This shows that the fluorescence quantum yield associated with the enol form is higher than in the case of the keto form.

It is generally accepted that in certain organic systems such as coumarins (derivatives of 1,2-benzopyrone) or 1,3,4-thiadiazoles, fluorescence quantum yield tends to be higher in polar solvents [24,63], which is exactly the reason why Stokes shift values are best analyzed relative to the solvent’s polarity function E_T_(30) (see Appendix A). Figure 11a illustrates such correlation and clearly indicates a dual linear relationship. In particular, non-polar solvents, e.g., *n*-heptane or *n*-hexane, are aligned along one line, while polar solvents are aligned along the other, with the two lines considerably distanced from one another. The ESIPT effect is commonly observed in the first group of solvents, even at low NTBD concentrations. In the case of polar solvents, the Stokes shift values increase with changing solvent polarity due to, among other factors, interactions associated with the formation of hydrogen bonds between molecules of the fluorophore and the solvent. Finally, the respective relationships between Φ_F_ and log(k_nr_/k_r_) and the solvent’s polarity function E_T_(30) are demonstrated in Figure 11b,c. There are evident similarities between the correlations presented in [44] and Figure 11b, which corroborate the relationship between fluorescence quantum yield and solvent polarity. Simultaneously, Figure 11c shows that in solvents in which the radiative constant for NTBD fluorescence transition was observed to increase, e.g., in ethanol or methanol, the more efficient, radiative channels of excitation deactivation became dominant. The results presented in Figure 11b illustrate that the quantum yield calculated for the NTBD molecule in solvents conducive to the keto* emission was very low, which fits the specific characteristics, particularly the shorter lifetime, of the keto* form (see Table 1).

As presented above and in [44], under the described experimental conditions, NTBD in non-polar solvents tends to produce dual emission (Figure 2b and Appendix A) composed of a higher-intensity low-energy signal and a lower-intensity high-energy one, as was the case in *n*-hexane; note that in non-polar solvents with increased polarizability, this intensity trend may be reversed, with the higher-energy emission prevailing over the lower-energy counterpart [44]. Relatively significant Stokes shifts were observed for the lowest-energy emission band, which suggests a significant change in the underlying emitting molecular state, consistent with the emergence of ESIPT. Moreover, higher-energy signals were visibly bathochromically shifted with the solvent’s increasing polarity and capacity to form hydrogen bonds (Figure 2b and Appendix A). This may be due to the fact that, as shown in [44] and Figure 11, polar (protic) solvents stabilize the underlying excited state to a greater extent than the ground state. As such, the bathochromic shift results from the reduction in the energy difference between the two states. At the same time, increasing solvent polarity triggered a hypsochromic shift in the lower-energy emission. This corroborates the observation that the corresponding excited state is less polar than the ground state, which is characteristic of keto* emission in ESIPT-capable molecules and further supports the assignment of higher-energy/lower-energy fluorescence signals to enol*/keto* tautomers, respectively.

## 3. Materials and Methods

### 3.1. Materials

NTBD synthesis was performed at the Department of Chemistry of the University of Life Sciences in Lublin. The process was described in the previous publication [43].

To prepare stock solutions of NTBD, 1 mg of the compound was dissolved in selected solvents (methanol, ethanol, butan-1-ol, propan-2-ol, acetonitrile, ethyl acetate, DMSO, DMF, THF, acetone, toluene, chloroform, *n-*hexane, and *n-*heptane, all of analytical grade). An adequate volume of the solution was measured and added to 2 mL of solvent to obtain the required absorbance intensity. The molar concentrations of the dissolved NTBD were 1.49 × 10^−3^ M in THF and 1.35 × 10^−3^ M in DMSO.

### 3.2. Methods

#### 3.2.1. General Experimental Techniques

A Cary 300 Bio double-beam UV–vis spectrophotometer (Varian, Palo Alto, CA, USA) was used to record electronic absorption spectra for NTBD. Fluorescence excitation, emission and synchronous spectra were all measured at 22 °C using a Cary Eclipse spectrofluorometer (Varian, Palo Alto, CA, USA). Resonance light scattering (RLS) measurements were performed in line with the protocol previously reported in [52]; the recorded data were analyzed with Grams/AI 8.0 software (Thermo Electron Corporation; Waltham, MA, USA). Quantum yields (Φ_F_) of fluorescence for NTBD were calculated relative to 7-diethylamino-4-methylcoumarin (coumarin 1) taken as standard ($\Phi_{F}=0.73$ in ethanol) [64]. Those values along with fluorescence lifetimes (τ) were used to determine rate constants for radiative and non-radiative processes [63,65]. Additional experimental details can be found in Appendix A.

#### 3.2.2. Fluorescence Lifetimes

The frequency domain method was employed to measure fluorescence lifetimes (τ) using a multifrequency cross-correlation phase and modulation K2 fluorometer (ISS, Champaign, IL, USA). NTBD was dissolved in an organic solvent or a mixture of methanol and water. In each sample, the fluorescence emission was recorded in a 10 × 10 mm quartz cuvette (4 × 4 mm for high NTBD concentrations) with excitation at 315 nm (300 W xenon arc lamp) using a cut-off filter (transmittance for λ > 320 nm) in the emission channel. The measurements were carried out at 15–20 modulation frequencies ranging from 2 to 200 MHz, with a water solution of Ludox@ (Aldrich, Darmstad, Germany) used for reference. Data analyses were conducted with the Vinci 2.0 software from ISS based on the multiexponential decay model for discrete fluorescence lifetime components shown in Equation (1):(1)Iλ,t=∑ifi (λ)tie−tti
where $I\left(\lambda ,t\right)$ is the fluorescence intensity and $f_{i}(\lambda )$ is the fractional contribution of each fluorescence lifetime component $t_{i}$.

Best-fit parameters were calculated by minimizing the reduced χ^2^ value and the residual distribution of the experimental data. Each measurement was taken 3–5 times, after which the obtained results were averaged and standard deviations calculated.

#### 3.2.3. Quantum Chemical Calculations

All calculations were performed using density functional theory (DFT) and its time-dependent variant (TDDFT) methods. Solvent effects for methanol (MeOH, ε = 32.613) and water (H_2_O, ε = 78.3553) were modeled by employing the polarizable continuum model (PCM) [66]. Systematic conformational analysis in the S_0_ ground state, including rotations around several bonds as indicated in Appendix A, was carried out using the B3LYP [67,68,69] density functional and the 6-311++G(d,p) [70,71] basis set. Dispersion effects were accounted for in these calculations via the third-generation Grimme’s set of semiempirical dispersion corrections with Becke–Johnson damping (D3) [72,73]. The absorption spectra and emission properties were modeled with B3LYP and the aug-cc-pVDZ [74,75] basis set via S_1_ excited-state geometry optimizations. Energy barriers for the enol ↔ keto tautomerization in both the S_0_ ground state and S_1_ excited state were determined based on the corresponding energy profile calculations for the proton transfer from the *cis*-enol structure to the respective keto performed with B3LYP/aug-cc-pVDZ. The same level of theory was also employed to study the possibility of intramolecular rotation of the resorcynyl fragment relative to the 1,3,4-thiadiazole unit in the S_1_ excited-state *cis*-enol and keto structures. A full description of the computational details used in the presented studies along with appropriate references is provided in the Appendix A.

## 4. Conclusions

As conclusively demonstrated and confirmed in the extensive experimental and computational studies presented above involving spectroscopic measurements acquired with several instrumental techniques as well as quantum chemical (time-dependent) density functional theory calculations, the effect of dual fluorescence emission observed for the NTBD resorcynyl–thiadiazole derivative is related to enol → keto phototautomerization processes occurring in S_1_ with the involvement of excited-state intramolecular proton transfer (ESIPT) pathways. As long as the *cis*-enol NTBD form is predominant in a solution, a single emission is observed; however, when both tautomers are present in S_1_ due to their co-existence in S_0_ or the efficient *cis*-enol* → keto* ESIPT process, a dual emission signal is induced, with its higher-energy/lower-energy band originating from the *cis*-enol*/keto* forms. After the emission, the keto form quickly isomerizes to the *cis*-enol state, which is more stable in S_0_. The presented studies also evidenced important cooperativity between the ESIPT and aggregation effects, which was particularly apparent in mixed media such as MeOH/H_2_O. While water molecules themselves can provide a crucial assistance for ESIPT, the (offset) “card pack” aggregation of NTBD observed in mixed solvent solutions ensures the co-planarity of resorcynyl and thiadiazole rings in the molecular structure of the compound due to restriction of the intramolecular rotation mechanism. This, in effect, prevents the formation of non-emissive TICT keto* states and therefore leads to considerably higher intensity of the lower-energy bands of keto* origin via aggregation-induced emission effects.

## Data Availability

The insight into detailed data might be obtained after the contact with the corresponding authors.

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
