# Peer review of "Cooperativity of ESPT and Aggregation-Induced Emission Effects—An Experimental and Theoretical Analysis of a 1,3,4-Thiadiazole Derivative"

_ijms, 2024, doi:10.3390/ijms25063352_

Round 1
Reviewer 1 Report
Comments and Suggestions for Authors
Manuscript reports on thorough study of photo-physical properties and solvent effects in the system of 1,3,4-thiadiazole derivate. The experimental and theoretical data are presented. Manuscript makes a solid contribution into the field, it is written in a good English. Nevertheless, I found it hard to read because to overwhelmed text volume and not optimal figure design. Therefore, I recommend shortening and reorganizing the manuscript. The following points should be addressed:
1. Scheme 1 shows the geometry of different forms of NTBD molecule, in fact the ‘NTBD’ in upper-left corner of scheme and ‘cis-enol’ are the same form of molecule, this repetition needs to be eliminated.
2. The experimental part should be shortened and partially transferred to supplementary materials. Namely, the sections 2.2.1, 2.2.2 and 2.2.3 seems to be better placed in SM in the present form, while short description of experimental technique should be in the main text instead.
3. Section 2.2.5 ‘Systematic conformational analysis in the S0 ground state, including rotations 180 around several bonds as indicated in Figure S7, was carried out using the B3LYP [50-52] 181 density functional and the 6‑311++G(d,p) [53, 54] basis set. ’, ‘The absorption spectra and emission properties via S1 excited-state geometry optimizations were modelled with B3LYP and the aug-cc-pVDZ [57, 58] basis set. ’. Please, justify the choice of B3LYP functional (but not more special functionals, for example, CAM-B3LYP, ωB97X-D or BMK). Explain why geometry optimization and property calculations were carried out using different series of basis sets.
4. Figure 1 shows too small molecular pictograms and small digits hard to read. Perhaps schematic representation will be more clear.
‘Numbers displayed along arrows are estimated energy barriers of the transition between two adjacent minima in the direction indicated by the arrow with respect to energy profiles connecting given structures. ’ There are two sets of numeric values above and below arrows, which are significantly different. What is the nature of these certain values?
5. Lines 232-233 ‘More importantly, in agreement with the experimental data [presented in this manuscript], the computational results confirm …’ Please, specify the certain ‘experimental data’ and make the references on the respective figures/tables.
Author Response
Reviewer 1:
Manuscript reports on thorough study of photo-physical properties and solvent effects in the system of 1,3,4-thiadiazole derivate. The experimental and theoretical data are presented. Manuscript makes a solid contribution into the field, it is written in a good English. Nevertheless, I found it hard to read because to overwhelmed text volume and not optimal figure design. Therefore, I recommend shortening and reorganizing the manuscript. The following points should be addressed:
- Scheme 1 shows the geometry of different forms of NTBD molecule, in fact the ‘NTBD’ in upper-left corner of scheme and ‘cis-enol’ are the same form of molecule, this repetition needs to be eliminated.
The requested change has been applied.
- The experimental part should be shortened and partially transferred to supplementary materials. Namely, the sections 2.2.1, 2.2.2 and 2.2.3 seems to be better placed in SM in the present form, while short description of experimental technique should be in the main text instead.
The requested change has been applied.
- Section 2.2.5 ‘Systematic conformational analysis in the S0 ground state, including rotations 180 around several bonds as indicated in Figure S7, was carried out using the B3LYP [50-52] 181 density functional and the 6‑311++G(d,p) [53, 54] basis set. ’, ‘The absorption spectra and emission properties via S1 excited-state geometry optimizations were modelled with B3LYP and the aug-cc-pVDZ [57, 58] basis set. ’. Please, justify the choice of B3LYP functional (but not more special functionals, for example, CAM-B3LYP, ωB97X-D or BMK). Explain why geometry optimization and property calculations were carried out using different series of basis sets.
The global hybrid B3LYP density functional was chosen for these studies due to its successful employment in quantum-chemical characterization of numerous 1,3,4-thiadiazole derivatives including resorcynyl-based ones [J. Lumin. 2018, 201, 44-56; J. Fluoresc. 2018, 28, 65-77; Int. J. Mol. Sci. 2019, 20, 5494; J. Mol. Liq. 2019, 291, 111261; Molecules 2020, 25, 4168; Molecules 2020, 25, 2822; Photochem. Photobiol. Sci. 2020, 19, 844-853]. While the correlation consistent polarized valence double-zeta diffuse-augmented (aug-cc-pVDZ) basis set was indicated in the aforementioned previous research [Int. J. Mol. Sci. 2019, 20, 5494; J. Mol. Liq. 2019, 291, 111261; Molecules 2020, 25, 4168] as providing a satisfactory agreement between the computed and experimental photophysical properties, for this project we initially chose a triple-zeta basis set with polarization and diffuse functions added, 6-311++G(d,p), of presumably better quality in terms of a description of valence electrons. Unfortunately, after the successful employment of B3LYP/6-311++G(d,p) for S0 conformational analysis, the preliminary TDDFT excited state S1 geometry optimizations showed that with such method no stable planar keto S1 structure can be found as all the calculations led to the twisted geometry instead. Consequently, photophysical properties were finally computed with the aug-cc-pVDZ basis set that demonstrated similar results to 6-311++G(d,p) for enol S1 but enabled to locate both planar and twisted keto S1 forms. This explanation has been added to the computational details described in the Supplementary Materials.
- Figure 1 shows too small molecular pictograms and small digits hard to read. Perhaps schematic representation will be more clear.
‘Numbers displayed along arrows are estimated energy barriers of the transition between two adjacent minima in the direction indicated by the arrow with respect to energy profiles connecting given structures. ’ There are two sets of numeric values above and below arrows, which are significantly different. What is the nature of these certain values?
Figure 1 has been rearranged to increase readability. Two sets of numerical values correspond to energy barriers for given forward and reverse transitions – this has been now clearly stated in the revised caption of the figure.
- Lines 232-233 ‘More importantly, in agreement with the experimental data [presented in this manuscript], the computational results confirm …’ Please, specify the certain ‘experimental data’ and make the references on the respective figures/tables.
The requested change has been applied by replacing ‘[presented in this manuscript]’ with ‘(see UV-vis absorption analysis below, Figures 2 and 3)’.

Reviewer 2 Report
Comments and Suggestions for Authors
The manuscript entitled “Cooperativity of ESPT and Aggregation-Induced Emission Effects – an Experimental and Theoretical Analysis of a 1,3,4-Thiadiazole Derivative” submitted by I. Budziak-Wieczorek and coworker presents a combined experimental and theoretical investigation about the photochemical behavior of 4-[5-(Naphthalen-1-ylmethyl)-1,3,4-Thiadiazol-2-yl]Benzene-1,3-Diol (NTBD) considering various solvents and solvent mixtures. The research topic explored in the manuscript looks of having a significant importance since it is a systematic and exhaustive investigation about the nature of the adsorption and emission properties of NTBD.
General observation:
- The nature of the different electronic excitations could be better illustrated by using Natural Transition orbital transitions instead of canonical orbital ones (see the Gaussian program under the keyword Pop=(SaveNTO) or https://gaussian.com/faq4/) since it greatly simplifies the process of characterizing these transitions (Also see 10.1039/D3CP04226J). What’s more, the CT-type transitions can be more clearly characterized.
- Following on from the previous idea, the CT transitions mentioned in line 297 should be discussed in a little more detail.
- Figure 1 is very crowded, perhaps it would be better to split it into “cis-enol 1” and “cis-enol 2” cases.
- I consider the possible aggregation presented in Scheme 1 (panel b) to be speculative, as no thorough theoretical calculation has been done.
- It would be useful to split the "Results and discussion" section into several thematic subsections. It will be much more easily understandable.
- Following the emission phenomenon presented by the authors in Figure 2b, when the ACN solvent was used, it can be observed that the emission of the system excited at 375 wavelength is different from that when it is excited to a higher excited state at 286 wavelengths. This certainly suggests that a molecule-to-solvent proton transfer can occur and the relaxation channels are also influenced by the nature of the solvent.
- It is also worth taking into account some unwanted effects that can occur when measuring emissions. For example, the Inner Filter Effects, see 10.1021/acsenergylett.9b01146, this can seriously modify the emission spectra at different concentration or solvent mixture.
- Was the pH effect of the solvents treated? Proton transfer can occur not only during the excitation, but also before it and forming NTBD-H+ or NTBD(=O-) where the proton is attached to one of the two nitrogens or is detaches from one of the OH fragments. Small shoulders (See Figure 2a) could come from these effects.
Overall, the manuscript could be suitable for publication in the IJMS journal, but it needs for further major revision.
Author Response
Reviewer 2:
The manuscript entitled “Cooperativity of ESPT and Aggregation-Induced Emission Effects – an Experimental and Theoretical Analysis of a 1,3,4-Thiadiazole Derivative” submitted by I. Budziak-Wieczorek and coworker presents a combined experimental and theoretical investigation about the photochemical behavior of 4-[5-(Naphthalen-1-ylmethyl)-1,3,4-Thiadiazol-2-yl]Benzene-1,3-Diol (NTBD) considering various solvents and solvent mixtures. The research topic explored in the manuscript looks of having a significant importance since it is a systematic and exhaustive investigation about the nature of the adsorption and emission properties of NTBD.
General observation:
- The nature of the different electronic excitations could be better illustrated by using Natural Transition orbital transitions instead of canonical orbital ones (see the Gaussian program under the keyword Pop=(SaveNTO) or https://gaussian.com/faq4/) since it greatly simplifies the process of characterizing these transitions (Also see 10.1039/D3CP04226J). What’s more, the CT-type transitions can be more clearly characterized.
- Following on from the previous idea, the CT transitions mentioned in line 297 should be discussed in a little more detail.
We thank the reviewer for the suggestion and provided reference (it has been added to the references list). We agree that natural transition orbitals (NTOs) can indeed be very helpful in obtaining a qualitative description of electronic excitations, especially in the cases when the excited state is described by numerous transitions within delocalized canonical molecular orbitals set. As presented in the manuscript (and now, as requested by the reviewer, elaborated in more details), the main (lower-energy) band observed at around 320 nm in the simulated spectrum of for example cis-enol 1 structure originates from S0→S1 (#1) and S0→S2 (#2) electronic excitations involving predominantly HOMO→LUMO (62% in #1, 32% in #2) and HOMO-1→LUMO (34% in #1, 65% in #2) transitions with the former (HOMO→LUMO) exhibiting -* character within the -electron system extending over the resorcynyl-thiadiazole, and the latter (HOMO-1→LUMO) corresponding to the naphthalene→resorcynyl-thiadiazole charge transfer (CT). As expected, the generated NTOs (see Figure 1 below) demonstrate exactly the same nature of those excitations but with mixed -* and CT contributions as occupied excited particle NTO spreads across the whole p-electron system of the molecule. Accordingly, we believe that in this specific case the description of the relevant excitations using the canonical MOs is more informative as it allows for a separation of electronic transitions with different nature.
Figure 1. Description of S0→S1 and S0→S2 excitations for cis-enol 1 using canonical MOs (top) and NTOs (bottom). Number listed are percentage contributions of a given orbitals pair into the excitations 1 & 2. Isosurfaces: ±0.04 au. Based on B3LYP/aug-cc-pVDZ/PCM(MeOH) calculations.
- Figure 1 is very crowded, perhaps it would be better to split it into “cis-enol 1” and “cis-enol 2” cases.
The requested change has been applied.
- I consider the possible aggregation presented in Scheme 1 (panel b) to be speculative, as no thorough theoretical calculation has been done.
We agree with the reviewer and in the revised version of the manuscript we have highlighted the hypothetical nature of these motifs in the Scheme caption.
- It would be useful to split the "Results and discussion" section into several thematic subsections. It will be much more easily understandable.
The requested change has been applied.
- Following the emission phenomenon presented by the authors in Figure 2b, when the ACN solvent was used, it can be observed that the emission of the system excited at 375 wavelength is different from that when it is excited to a higher excited state at 286 wavelengths. This certainly suggests that a molecule-to-solvent proton transfer can occur and the relaxation channels are also influenced by the nature of the solvent.
We thank the Reviewer also for this observation with which we fully agree. Indeed, the emission spectra for NTBD in ACN noticeably varied depending on the excitation wavelength, i.e. 286 or 322. One of the possible explanations of this fact is that molecule-to-solvent proton transfer may have occurred, as rightly suggested by the Reviewer. At the same time, relaxation channels most certainly depend on the polarity of the medium used. In line with the Reviewer’s suggestion, we decided to add this information to the main body of the manuscript. In future research, we will attempt to investigate this matter in greater detail. The following sentence was added in the manuscript.
- It is also worth taking into account some unwanted effects that can occur when measuring emissions. For example, the Inner Filter Effects, see 10.1021/acsenergylett.9b01146, this can seriously modify the emission spectra at different concentration or solvent mixture.
We also thank the Reviewer for the above valuable comment that is very relevant to spectroscopic measurements, especially when it comes to fluorescence emission spectra. Inner Filter Effects may significantly influence the shape of such spectra, particularly at higher concentrations of the analyzed compounds. It should be noted, however, that when conducting our research we remained aware of this fact and were very careful to select suitable concentrations with this in mind.
As suggested by the Reviewer, we decided to mention this fact in the text and include the publication mentioned in the references as it is an overall very valuable source. The relevant change was made in the manuscript and the References section.
“…in order to avoid unwanted interference related to Inner Filter Effects [76 ])”
We added this publication to our Reference:
- Lewandowska-Andralojc, A., and Marciniak, B., Five major sins in fluorescence spectroscopy of light-harvesting hybrid materials. ACS Energy Letters, 2019. 4(8).
- Was the pH effect of the solvents treated? Proton transfer can occur not only during the excitation, but also before it and forming NTBD-H+ or NTBD(=O-) where the proton is attached to one of the two nitrogens or is detaches from one of the OH fragments. Small shoulders (See Figure 2a) could come from these effects.
We also thank the Reviewer for this valuable comment. We are of course aware of this effect and have in fact discussed it in our earlier publications focusing on other 1,3,4-thiadiazole analogues in water solutions. In the context of solvents, we intend to explore this problem in our future publications, including those related to model biological systems such as micelles or liposomal systems. Given the already considerable volume of the present paper, already at the initial stage of its preparation we decided not to include this particular problem. We are nonetheless very aware of the issue mentioned by the Reviewer. However, following the Reviewer’s suggestion, we intend to either explore it in a standalone publication, or include related considerations in a paper discussing the biological systems mentioned above.

Round 2
Reviewer 2 Report
Comments and Suggestions for Authors
This new revised version of the manuscript shows significant improvements compared to the first submitted version of the manuscript. The author has given satisfactory solutions and explanations for the addressed questions related to his original work.
The manuscript can be considered for publishing in IJMS in the present form.